# An innovative method to calibrate a spinner anemometer without use of yaw position sensor

Giorgio Demurtas[1] and Nick Gerardus Cornelis Janssen[2]

[1]DTU Wind Energy, Frederiksborvej 399, 4000 Roskilde, Denmark.
[2]Romo Wind A/S, Olof Palmes Alle 47, 8200 Aarhus N, Denmark.

*Correspondence to:* Giorgio Demurtas (giod@dtu.dk)

**Abstract.** A spinner anemometer can be used to measure the yaw misalignment and flow inclination experienced by a wind turbine. Previous calibration methods used to calibrate a spinner anemometer for flow angle measurements were based on measurements of a spinner anemometer with default settings (arbitrary values, generally $k_{1,d} = 1$ and $k_{2,d} = 1$) and a reference yaw misalignment signal measured with a yaw position sensor. The yaw position sensor is normally present in wind turbines for control purposes, however, such a signal is not always available for a spinner anemometer calibration. Therefore, an additional yaw position sensor was installed prior to the spinner anemometer calibration. An innovative method to calibrate the spinner anemometer without a yaw positions sensor was then developed. It was noted that a non calibrated spinner anemometer that overestimate (underestimate) the inflow angle will also overestimate (underestimate) the wind speed when there is a yaw misalignment. The new method leverage on the non linearity of the spinner anemometer algorithm to find the calibration factor $F_\alpha$ by an optimization process that minimizes the dependency of the wind speed to the yaw misalignment. The new calibration method was found to be rather robust with $F_\alpha$ values within $\pm 2.7\%$ of the mean value for four successive tests at the same rotor position.

## Nomenclature

| | | | |
|---|---|---|---|
| $V_1$ | Speed along the sensor path of probe 1. | $\alpha$ | Inflow angle respect to the shaft axis. |
| $V_2$ | Speed along the sensor path of probe 2. | $\delta$ | Shaft tilt angle. |
| $V_3$ | Speed along the sensor path of probe 3. | $\beta$ | Flow inclination angle. |
| $V_{ave}$ | Mean value between $V_1$, $V_2$, $V_3$. | $\gamma$ | Yaw misalignment. |
| $U$ | Wind speed vector modulus. | $\gamma_{ref}$ | Reference yaw misalignment. |
| $U_{hor}$ | Horizontal wind speed component. | $\phi$ | Rotor position |
| $U_{hor,d}$ | Horizontal wind speed (non calibrated). | $\overline{U_{hor}}$ | Mean horizontal wind speed. |
| $U_{hor,d,c}$ | Horizontal wind speed component (calibrated with correct $k_\alpha$ but not yet $k_1$). | $\theta$ | Azimuth position of flow stagnation point on spinner (relative to sonic sensor 1). |

| $k_1$ | Calibration constant mainly related to wind speed calibration. | $F_1$ | Calibration correction factor mainly related to wind speed calibration. |
|---|---|---|---|
| $k_\alpha$ | Calibration constant mainly related to angle calibration. | $F_\alpha$ | Calibration correction factor mainly related to angle calibration. |
| $k_2$ | Calibration constant (equal to $k_\alpha \cdot k_1$). | $F_2$ | Calibration correction factor ($F_\alpha \cdot F_1$). |
| RMSE | Root Mean Square Error | TI | Turbulence Intensity. |
| QSC | Quality Score | WSR | Wind Speed Response method |
| GGref | Gamma-Gamma reference method | TanTan | Tangent-Tangent method |

## 1 Introduction

The spinner anemometer (Pedersen et al. (2007)) measures the horizontal wind speed $U_{hor}$, yaw misalignment $\gamma$ and flow inclination $\beta$ experienced by a wind turbine by measuring the flow on the spinner by using three 1D sonic sensors. The three
1D sonic sensors are mounted on the spinner and connected to a so called "conversion box". Each sonic sensor arm also contains a 1D accelerometer which measurements are used in the conversion box to calculate the rotor position. The main purpose of the conversion box is to execute the conversion algorithm that transform the 1D sonic sensors readings which are in a rotating coordinate reference system (Fig. 1) to the fixed nacelle coordinate reference system as $U_{hor}$, $\gamma$ and $\beta$. The conversion algorithm takes into consideration the wind turbine tilt angle $\delta$ which is set in the conversion box as a constant.
The shape of the spinner is accounted for by two calibration coefficients, $k_1$ and $k_2$. The first coefficient mainly relates to wind speed measurements, while the ratio of the two coefficients $k_\alpha = k_2/k_1$ mainly relates to flow angle measurements. The relations between the wind speed $U$, flow angle $\alpha$ and azimuth position of the stagnation point $\theta$ producing $V_1$, $V_2$ and $V_3$ measured by the three 1D sonic sensors are:

$$V_1 = U\left(k_1\cos(\alpha) - k_2\sin(\alpha)\cos(\theta)\right) = U \cdot k_1\left(\cos(\alpha) - k_\alpha\sin(\alpha)\cos(\theta)\right) \tag{1}$$

$$V_2 = U\left(k_1\cos(\alpha) - k_2\sin(\alpha)\cos\left(\theta - \frac{2\pi}{3}\right)\right) = U \cdot k_1\left(\cos(\alpha) - k_\alpha\sin(\alpha)\cos\left(\theta - \frac{2\pi}{3}\right)\right) \tag{2}$$

$$V_3 = U\left(k_1\cos(\alpha) - k_2\sin(\alpha)\cos\left(\theta - \frac{4\pi}{3}\right)\right) = U \cdot k_1\left(\cos(\alpha) - k_\alpha\sin(\alpha)\cos\left(\theta - \frac{4\pi}{3}\right)\right) \tag{3}$$

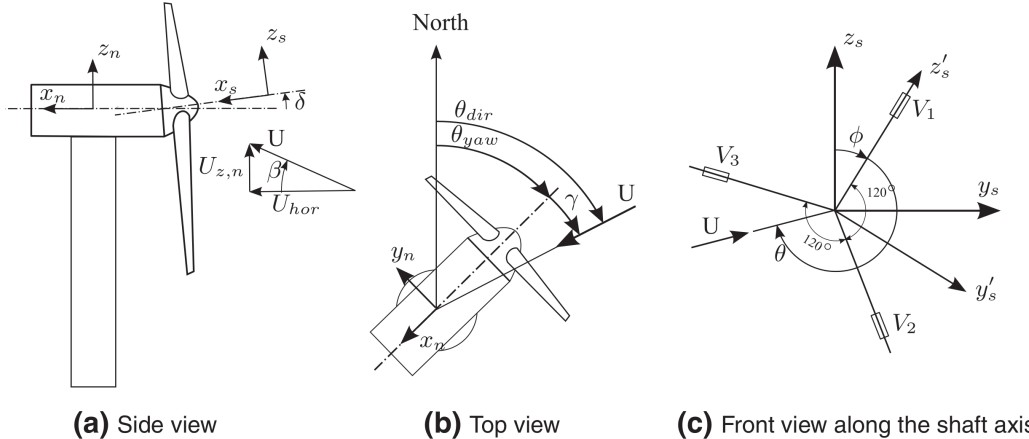

**(a)** Side view      **(b)** Top view      **(c)** Front view along the shaft axis

**Figure 1.** Coordinate systems and definition of angles: rotating spinner coordinate system $x'_s$, $y'_s$ and $z'_s$, non-rotating shaft coordinate system $x_s$, $y_s$ and $z_s$, fixed nacelle coordinate system $x_n$, $y_n$ and $z_n$, yaw direction $\theta_{yaw}$, yaw misalignment $\gamma$, flow inclination angle $\beta$, tilt angle $\delta$, azimuth position of flow stagnation point on spinner $\theta$ (relative to sonic sensor 1) and rotor azimuth position $\phi$ (position of sonic sensor 1 relative to vertical). From Demurtas et al. (2016).

The conversion algorithm (Eq. 5 to Eq. 18) was derived from Eq. 1, 2 and 3. The values of $k_1$ and $k_2$ constants are generally not know when the spinner anemometer is installed on a wind turbine for the first time, they are therefore set to an arbitrary value, generally $k_{1,d} = 1$ and $k_{2,d} = 1$. The calibration procedure will then provide the correction factors $F_1$ and $F_\alpha$ to correct the default values to calibrated values (Eq. 4). The output values relative to a spinner anemometer which is measuring with

5    default calibration settings has the subscript $'d'$ ($U_{hor,d}$, $\gamma_d$, $\beta_d$ ).

$$k_1 = F_1 \cdot k_{1,d} \qquad\qquad k_2 = F_2 \cdot k_{2,d} = k_\alpha \cdot k_1 = k_{\alpha,d} \cdot F_\alpha \cdot k_1 \qquad\qquad (4)$$

$$\alpha = \arctan\left(\frac{k_1\sqrt{3(V_1 - V_{ave})^2 + (V_2 - V_3)^2}}{\sqrt{3}k_2 V_{ave}}\right) \qquad\qquad (5)$$

$$V_{ave} = \frac{1}{3}(V_1 + V_2 + V_3) \qquad\qquad (6)$$

$$U = \frac{V_{ave}}{k_1 \cos\alpha} \qquad\qquad (7)$$

10    $$V_1 < V_{ave} : \theta = \arctan\frac{(V_2 - V_3)}{\sqrt{3}(V_1 - V_{ave})} \qquad\qquad V_1 \geq V_{ave} : \theta = \arctan\frac{(V_2 - V_3)}{\sqrt{3}(V_1 - V_{ave})} + \pi \qquad\qquad (8)$$

$$U_{x,s} = U\cos(\alpha) \tag{9}$$

$$U_\alpha = U\sin(\alpha) \tag{10}$$

$$U_{y,s} = -U_\alpha \sin(\phi + \theta) \tag{11}$$

$$U_{z,s} = -U_\alpha \cos(\phi + \theta) \tag{12}$$

$$U_x = U_{x,s}\cos(\delta) + U_{z,s}\sin(\delta) \tag{13}$$

$$U_y = U_{y,s} \tag{14}$$

$$U_z = U_{z,s}\cos(\delta) - U_{x,s}\sin(\delta) \tag{15}$$

$$U_{hor} = \sqrt{U_x^2 + U_y^2} \tag{16}$$

$$\gamma = \arctan\left(\frac{U_y}{U_x}\right) \tag{17}$$

$$\beta = \arctan\left(\frac{U_z}{U_{hor}}\right) \tag{18}$$

## 1.1 Existing calibration methods for flow angle measurements

Two methods based on measurements to calibrate a spinner anemometer for flow angle measurements proposed in Pedersen and Demurtas (2014) consist in yawing the wind turbine of $\pm\ 60°$ several times under manual control (as indicated by the turbine yaw position sensor, with respect to the mean wind direction). During this test, the output parameters of the spinner anemometer ($U_{hor}$, $\gamma$, $\beta$) are recorded at high sampling frequency (10 Hz). The analysis of the measurements provide the correction factor $F_\alpha$ that multiplied by the default $k_{\alpha,d}$ gives the correct $k_\alpha$ calibration value.

The methods are based on the assumption that the wind direction is constant during the test. Due to this requirement, Pedersen and Demurtas (2014) recommended to do the test at wind speeds above 6 m/s. Both methods need the yaw position to be measured in order to calculate the reference yaw misalignment $\gamma_{ref}$, defined as the mean wind direction minus the instantaneous yaw position during test (see Pedersen and Demurtas (2014) for details). In the first method (abbreviated as GGref) $F_\alpha$ was calculated by calibrating the measurements iteratively, until the linear fit of $\gamma$ as a function of $\gamma_{ref}$ was giving a line of slope equal to 1.

In the second method (abbreviated as TanTan), only one linear fitting was made to $\tan(\gamma)$ as a function of $\tan(\gamma_{ref})$. In this case, the slope coefficient of the fit was exactly $F_\alpha$. The two calibration methods were found to be sensitive to the width of the yawing span. In fact, different $F_\alpha$ values were obtained sub-setting the data-set to a variable span of $\gamma_{ref}$.

A new method to find the $F_\alpha$ value, that does not require a yaw position measurement, and use the non linearity of the spinner anemometer conversion algorithm is proposed.

## 2 The wind speed response method

The method (abbreviated WSR) is based on the assumption that the wind speed is constant during the test. The turbulence of the real wind will add some scatter in the measurements which will reduce the repeatability of the result. While in principle a single yawing movement is sufficient, in practice the wind speed fluctuations needs to be averaged by yawing the wind turbine several times. The spinner anemometer is able to measure inflow angles (yaw misalignment $\gamma$ and flow inclination $\beta$) and wind speed $U$. A wrong $k_\alpha$ value will result in a wrong value of the angle $\gamma$, which will turn into a wrong value of the horizontal wind speed $U_{hor}$. In other words, a wrong $k_\alpha$ makes the wind speed measurement dependent on the yaw misalignment. This property of the spinner anemometer model (Eq. 1, 2 and 3) was verified with a data-set consisting of constant wind speed $U_{hor}$ and 13 values of yaw misalignment going from -60° to 60° in steps of 10°. The tilt angle and the flow inclination were set to arbitrary values (equal to zero for Fig. 2). In the real world the tilt angle of the wind turbine is typically between 3° and 6° while the flow inclination varies within approximately$\pm10°$. The conversion algorithm takes into consideration both the tilt angle $\delta$ and the measured flow inclination $\beta_d$ when calculating the yaw misalignment $\gamma_d$, therefore they have no influence on the result of this method. $V_1$, $V_2$ and $V_3$ were calculated with Eq. 1, 2 and 3 with $k_\alpha = 1$ and $k_2 = 1$.

Eq. 5 to Eq. 18 (which are the direct conversion algorithm presented in Pedersen and Demurtas (2014)) were used with new values of $k_\alpha$ equal to 0.5, 1 and 2, with the calculated $V_1$, $V_2$ and $V_3$ to calculate $U_{hor,d}$ and $\alpha_d$. $k_1$ was kept equal to one.

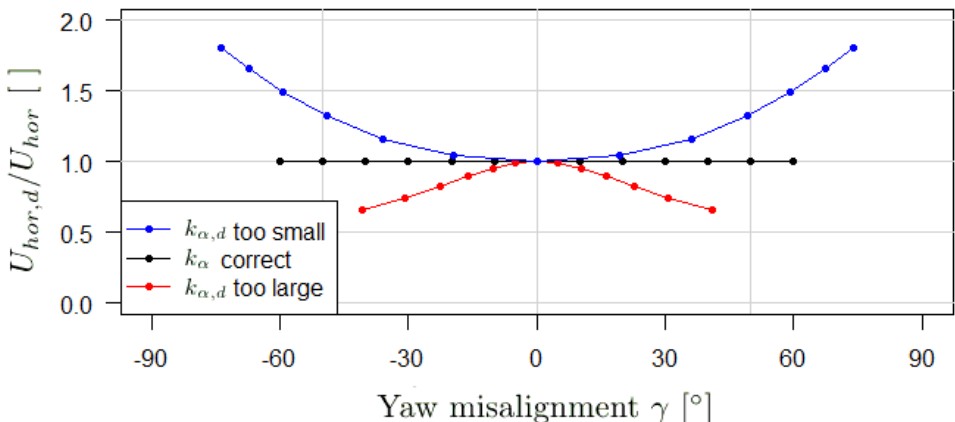

**Figure 2.** Effect of three $k_\alpha$ values on yaw misalignment and wind speed measurements. Black line shows data where the $k_\alpha$ is correct (equal to one for our theoretical spinner model). Blue curve shows $k_\alpha$ set to 0.5. To correct the blue curve to the black curve, the correction should be made with $F_\alpha > 1$ ($F_\alpha = 2$ in this case). Red line shows $k_\alpha$ set to twice the correct value, therefore we need $F_\alpha < 1$ to correct the measurements to the black line.

When the conversion was made with $k_\alpha = 1$, $U_{hor,d}$ and $\alpha_d$ matched the (correct) initial values of $U_{hor}$ and $\alpha$ (black line in Fig. 2). On the other hand, when the conversion was made with $k_{\alpha,d} = 0.5$ the wind speed and angle were overestimated (blue curve in Fig. 2) because $k_{\alpha,d}$ is too small compared to the correct $k_\alpha$ value equal to one in this example. Similarly, with $k_{\alpha,d} = 2$ the angles and the wind speed were underestimated (red curve in Fig. 2).

From experience of calibration on several turbines the default settings of $k_{\alpha,d} = 1$ is too small. Therefore the wind speed response looks like a happy smile and an $F_\alpha > 1$ is required to correct the default calibration value. Note that the wind speed is still measured correctly for small inflow angle (where the three curves of Fig. 2 are close to each other).

The method to optimize $F_\alpha$ consists in minimizing the RMSE (Root Mean Square Error) of a horizontal linear fit made to the measurements of $U_{hor,d}$ for varying $F_\alpha$. $U_{hor}$ is obtained applying the $F_\alpha$ calibration to the measurements of $U_{hor,d}$, $\gamma$, $d$

and $\beta_d$ acquired with default values $k_{1,d}$, $k_{2,d}$. For this reason $U_{hor}$ is a function of $U_{hor,d}$, $\gamma_d$, $\beta_d$, $k_{1,d}$, $k_{2,d}$ and $F_\alpha$.

The function object of the optimization is

$$RMSE = f(U_{hor,d}, \gamma_d, \beta_d, k_{1,d}, k_{2,d}, F_\alpha) = \sqrt{\frac{1}{n} \sum_1^n (\overline{U_{hor}} - U_{hor})^2}, \tag{19}$$

where the first three variables comes from the measurements, fourth and fifth are the settings of the spinner anemometer at the time of acquisition of the measurements, and the last one ($F_\alpha$) is the independent variable used in the optimization. The

function of Eq. 19 was optimized to its minimum using a combination of golden section search and successive parabolic interpolation (Brent and N.J. (1973)).

## 3 Application of the method

The measurements were acquired in February 2016 on a Neg-Micon 2 MW wind turbine installed in Denmark. The wind turbine was yawed in and out of the wind several times with the rotor stopped with one blade pointing downwards. Figures 3 and 4 show the 10 Hz data recorded during the calibration procedure. Figure 3A, B and C show non calibrated measurements, while Fig. 4A, B and C show calibrated measurements. In both Fig. 3 and 4 the sub-figure A shows the time series of the yaw misalignment and yaw misalignment reference (measured with a yaw position sensor). Sub-figure B shows the time series of the wind speed. Sub-figure C shows the wind speed response as a function of yaw misalignment.

Figure 3D shows the value of $F_\alpha$ calculated with the three different methods (GGref and TanTan from Pedersen and Demurtas (2014) and the present method, WSR), for varying range of yawing the wind turbine out of the wind (data were filtered according to $\gamma_{ref}$ in steps of 5° span per side). The $F_\alpha$ value was calculated with the WSR method only if there were at least 30 seconds of measurements in the outmost 5° of the considered range (which justifies the fact that the scatter plot of Fig. 4C appears wider than the maximum range shown in Fig. 3D by the green line).

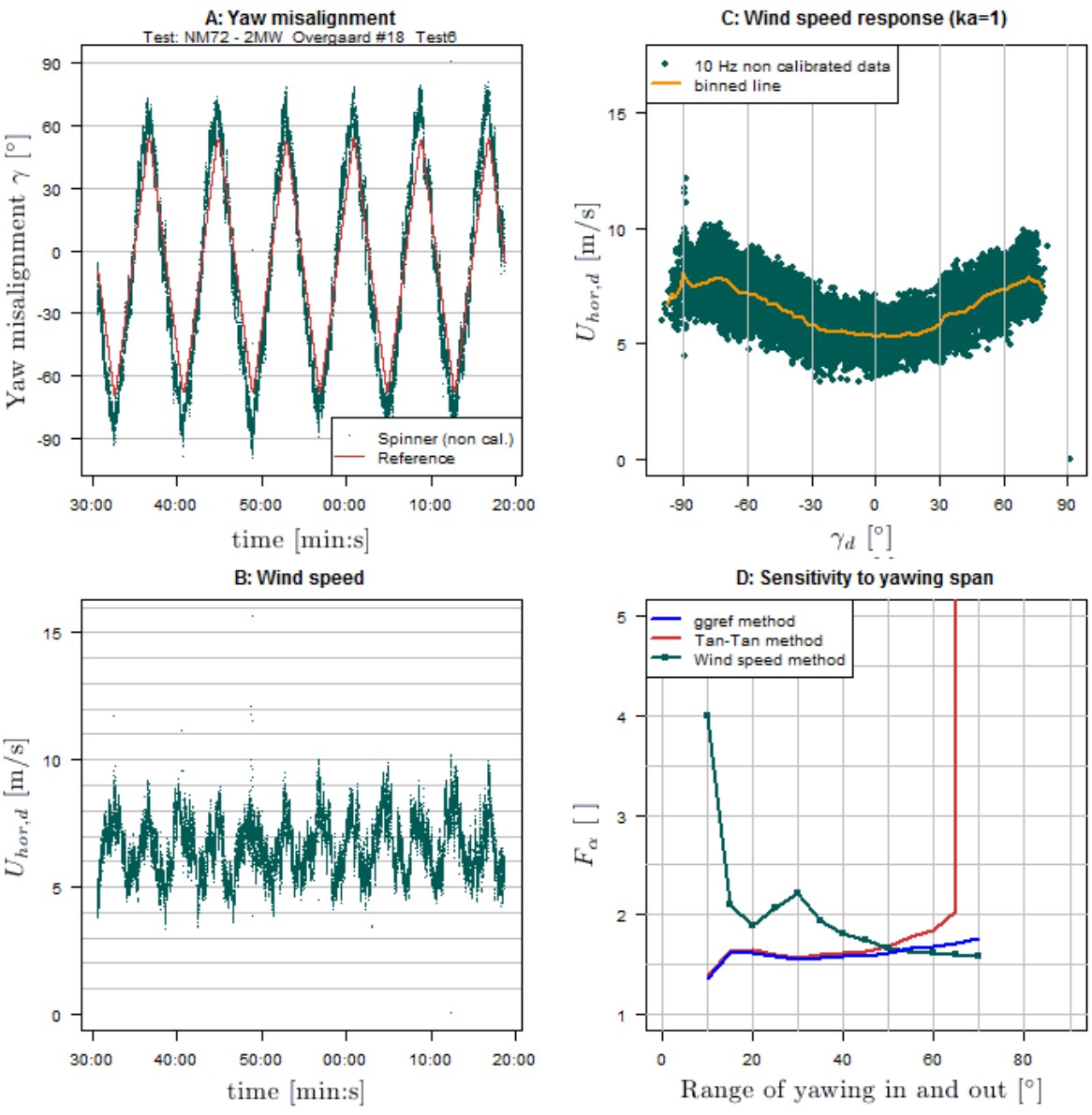

**Figure 3.** Before calibration, test 6. A: Time series of yaw misalignment as measured by the spinner anemometer and by the yaw position sensor. B: wind speed time series as measured by the spinner anemometer before F1 calibration. C: Wind speed as a function of yaw misalignment both measured by spinner anemometer. D: Calibration correction factor $F_\alpha$ calculated in three different methods, as a function of yawing span ranging from $\pm 10°$ to $\pm 90°$ in steps of $\pm 5°$.

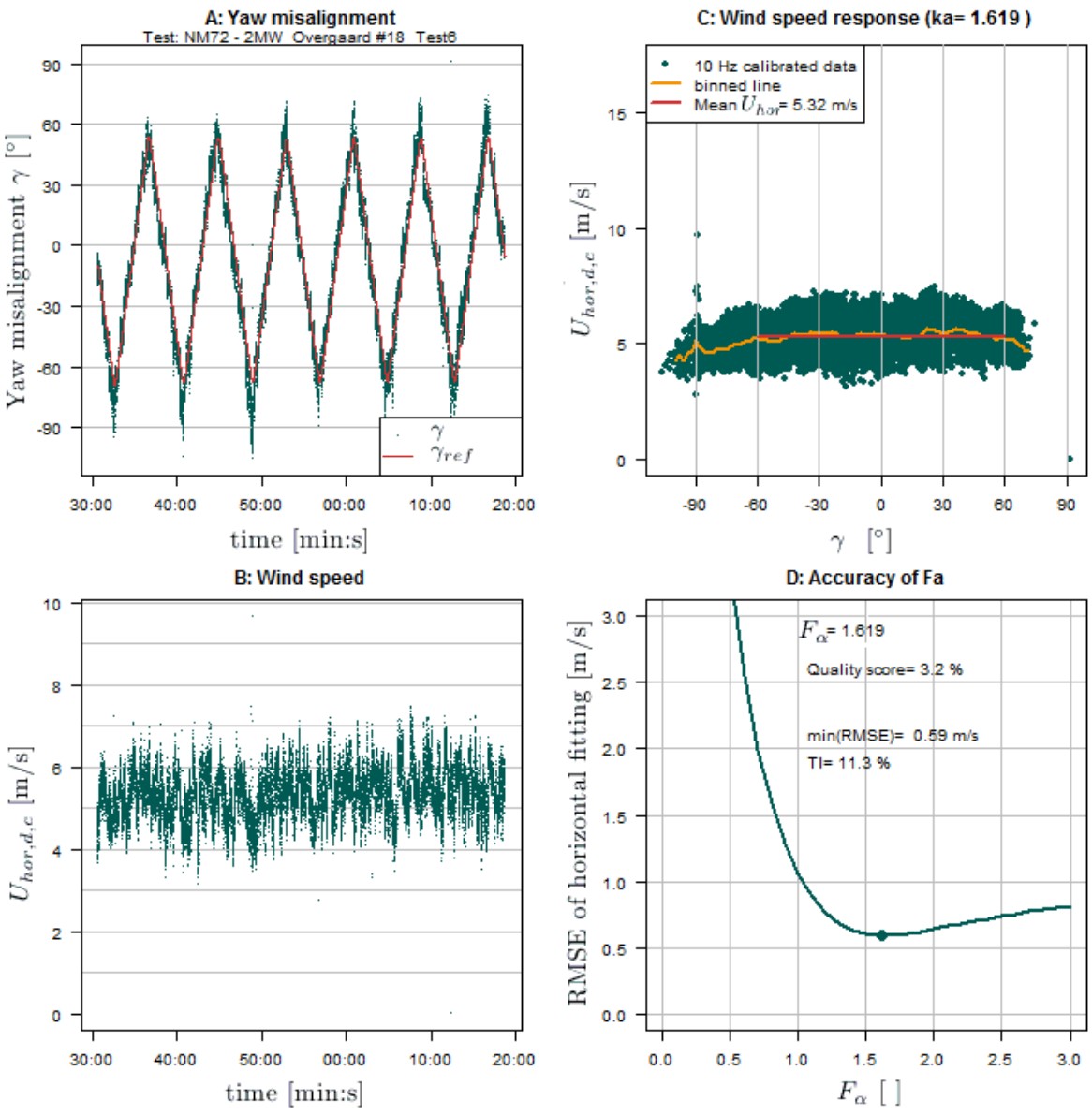

**Figure 4.** After calibration, test 6. A: Time series of yaw misalignment as measured by the spinner anemometer and by the yaw position sensor. B: wind speed time series as measured by the spinner anemometer before F1 calibration, after $F_\alpha$ calibration. C: Wind speed as a function of yaw misalignment both measured by spinner anemometer and calibrated with $F_\alpha$. D: Root mean square error of the horizontal fit (red line in sub-figure C) as a function of $F_\alpha$.

## 4 Discussion

As seen also in tests performed on other wind turbine models, the GGref and TanTan methods tend to give a higher $F_\alpha$ for increasing yawing span than the WSR method. This is especially true for the TanTan method, because of the tangent function properties, that tend to increase rapidly when approaching 90° angle.

As seen in Fig. 3D, the value of $F_\alpha$ is dependent on the chosen width of yawing the turbine in and out of the wind. For the TanTan and GGref methods, Pedersen and Demurtas (2014) suggested to limit the span to $\pm45°$. The value of $F_\alpha$ calculated with the WSR method tends to stabilize and be comparable with the previous two methods for a yawing span within 50° and 70°.

Above a certain large inflow angle (depending on the spinner shape) the air flow would separate from the spinner surface with the consequence of the downwind sensor measuring in a separated flow region. In this condition the spinner anemometer cannot measure correctly, since the relation between the sensor path velocities does not follow the spinner anemometer mathematical model (Eq. 1 to 3).

The $F_\alpha$ value calculated for yawing span of $\pm60°$ was 1.619. This value was used to calibrate the measurements, which are show in Fig. 4A, B and C. In Fig. 4C, the red line shows the mean wind speed for the measurements where the yawing span is in the range $\pm60°$. Figure 4D shows how the RMSE varies as a function of $F_\alpha$, and the optimum $F_\alpha$ with a dot at the minimum RMSE.

The method is based on the assumption of a constant wind speed. When applying the method to a spinner anemometer exposed to natural wind the wind speed will naturally vary in the time frame of about one hour needed to complete the six yawing cycles (Fig. 3A). The wind speed variations are clearly visible in the wide scatter of Fig. 3C, which are averaged when calculating the RMSE (Eq. 19). The turbulence reduces the repeatability of the result ($F_\alpha$) since it introduces some randomness in the measurements. The result can be improved by a large number of tests or by using a stable wind source. A worst case is that the increase (and decrease) of wind speed is synchronized with the yaw position of the turbine, which is basically impossible to happen when the turbine is yawed several times.

## 5 Sensitivity analysis

The calibration test was performed several times on the exact same turbine. The rotor was stopped with one blade pointing downwards (so called bunny position) and the nacelle was yawed six times for each test, of $\pm90°$ (test 7 to 10) or $\pm60°$ (test 1 to 6) by operating manually from the turbine control panel. The yaw moves with a speed of about 0.5°/s, therefore one test of six sweeps takes approximately one hour. Tests 7 to 10 were made in the same day one after the other for the exact same rotor position. The WSR method was used to calculate $F_\alpha$ for each test and several yawing span (Fig. 6) also reported in Tab. 1 for the case of $\pm 60°$ yawing span. Test 3 and 5 faced some data acquisition problems and were discarded.

Regarding the ability of the method to give reproducible results, the variation of $F_\alpha$ for tests 7 to 10 are within $\pm2.7\%$ of the mean value 1.52. Since the rotor position is the same for the four tests, the only ascribable responsible for the variations is the wind turbulence. The 8 results are within $\pm 8.5\%$ of the mean value 1.59. It seems that the $F_\alpha$ value relative to the first 4

tests (about 1.67) is higher than the last for tests (1.50), which could be due to a different rotor position which plays a role if the rotational symmetry of the spinner and sensor mounting positions are not accurate. The accuracy of the mounting position of the sonic sensors on this spinner was not investigated.

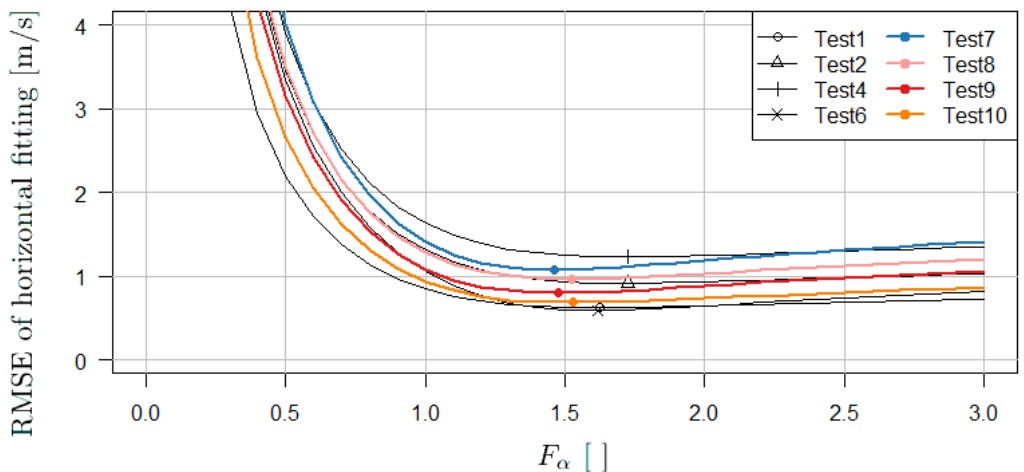

**Figure 5.** Root mean square error as a function of $F_\alpha$. Markers locate the minimum value of RMSE and the corresponding $F_\alpha$ value. Colour bold lines are tests performed for the exact same rotor position.

**Table 1.** $F_\alpha$ values for eight calibration tests made on the same wind turbine. Tests 7 to 10 were made with exact same rotor position relative to a wind turbine yawing span of $\pm 60°$.

| Test | 1 | 2 | 4 | 6 | 7 | 8 | 9 | 10 |
|---|---|---|---|---|---|---|---|---|
| $F_\alpha$ value | 1.63 | 1.72 | 1.73 | 1.62 | 1.46 | 1.53 | 1.48 | 1.53 |

## 6 Goodness of a calibration and benchmark on 17 wind turbine models

5    The variations encountered in the estimation of $F_\alpha$ call for the definition of a variable to judge the quality of the calibration. One indicator could be related to the shape of the curves of Fig. 5. The more flat and shallow minimum, the larger uncertainty on $F_\alpha$. The indicator was named quality score (QSC, see Eq. 20), calculated as the slope to the left of the minimum point.

$$QSC = \frac{RMSE(F_\alpha - 0.1) - RMSE(F_\alpha)}{0.1} \tag{20}$$

Figure 7 shows QSC as a function of the span of yawing.

10    What minimum quality score should a test have to give meaningful $F_\alpha$? To answer this question, the wind speed response method was applied to a database of yawing tests consisting of 29 calibration tests made on 17 turbine models. Results are shown in Fig. 8 and Fig. 9.

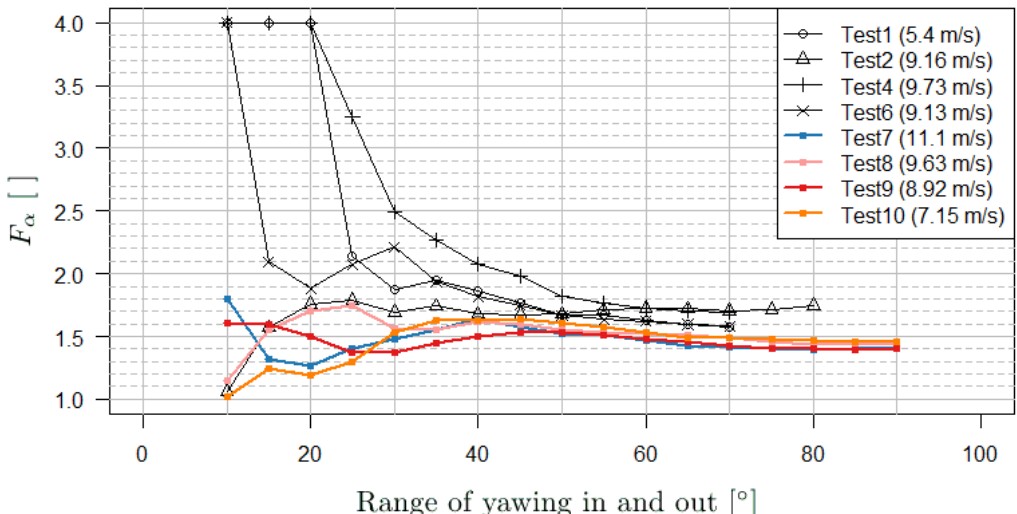

**Figure 6.** Sensitivity of the $F_\alpha$ to the yawing span. Colour bold lines are tests performed for the exact same rotor position. For test 2 the wind turbine was yawed of $\pm 60°$ but an initial offset of the turbine with respect to the wind direction and a wind direction change during the test determined measurements up to $80°$. The values in the legend shows the mean wind speed during the test.

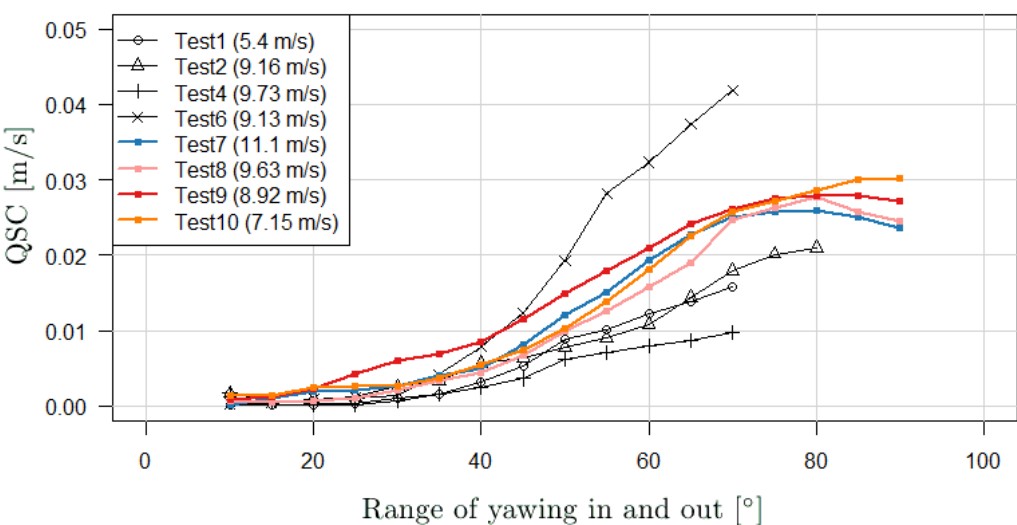

**Figure 7.** The quality score (QSC) is a measure of how much the RMSE as a function of $F_\alpha$ peaks at the minimum. A wide yawing span gives a more clear peak. The values in the legend shows the mean wind speed during the test.

Figure 8 can help to identify which conditions of wind speed and turbulence leads to a more precise estimate of $F_\alpha$, which means a more steep RMSE($F_\alpha$) curve, or in other words a high QSC. Average wind speed and turbulence intensity were calculated from the measurements calibrated with $F_\alpha$, for a range of yaw misalignments included in the interval $-30°$ to $30°$.

This is to ensure that there is not flow separation from the spinner surface and therefore ensure the spinner anemometer model validity (the spinner anemometer model is expressed by Eq. 1, Eq. 2 and Eq. 3). Figure 8 shows an inverse relation between the quality score and the turbulence intensity of the wind speed as measured by the spinner anemometer during the yawing test.

5    Figure 8 shows that the QSC increases with the wind speed $U_{hor}$.

The most pronounced correlation in Fig. 8 is between QSC and TI, where the QSC is increasing for decreasing turbulence intensity. This suggests that the ideal condition to perform the test is at low turbulence. The initial statement (in Sec. 2) that the wind speed turbulence would reduce the accuracy of the method is also confirmed by a QSC that reduces as the TI increases. A condition of low turbulent wind can practically be found by night, when the atmosphere is stable, in a side that is flat with

10    low roughness. It seems also that the QSC increases for increasing $U_{hor}$, however the scatter of QSC also increases and there are several points with a low QSC despite the high wind speed. This means that to achieve a high QSC it is more important a low TI than a high wind speed.

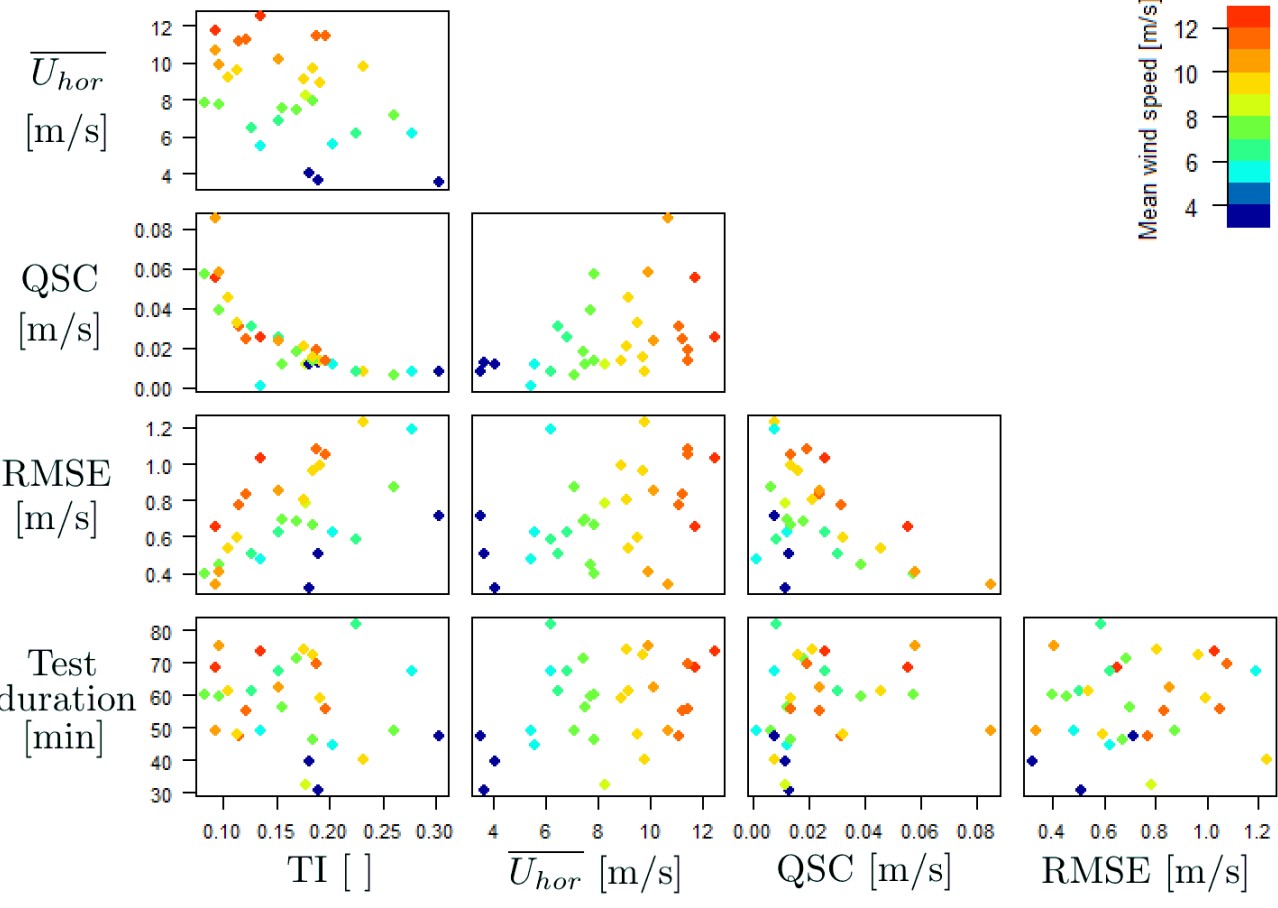

**Figure 8.** Application of the method to a large database of wind turbines. Colour coded with the mean wind speed.

## 7 Comparison with previous methods

The $F_\alpha$ was calculated with the three methods GGref, TanTan, and WSR for a range of yawing ($\gamma_{ref}$) of $\pm45°$, $\pm45°$ and $\pm60°$, respectively. Figure 9 shows a comparison of $F_\alpha$ values for 29 tests made on 17 wind turbine models. All the spinner anemometer were initially set with the same default calibration values ($k_{1,d} = 1$, $k_{2,d} = 1$) therefore it is possible to compare directly the $F_\alpha$ values. Most of the turbines present a $F_\alpha$ between 1 and 2, values which are attributable to a pointed spinner shape (like a Vestas V52) or a rounded spinner (like a Neg-Micon NM80). The four tests with $F_\alpha$ between 2.5 and 3.5 belong to a flat spinner like the one of a Siemens SWT-6.0-154.

The two methods which agrees the most are the GGref and the TanTan methods. This good agreement however does not implies that the $F_\alpha$ estimate is accurate, but rather that the two methods are similar (in fact they are both based on a linear fitting of the measurements, as described in Sec. 1.1).

The value of $F_\alpha$ calculated with the WSR method shows a lower level of agreement with the other two methods, being it based on a completely different principle.

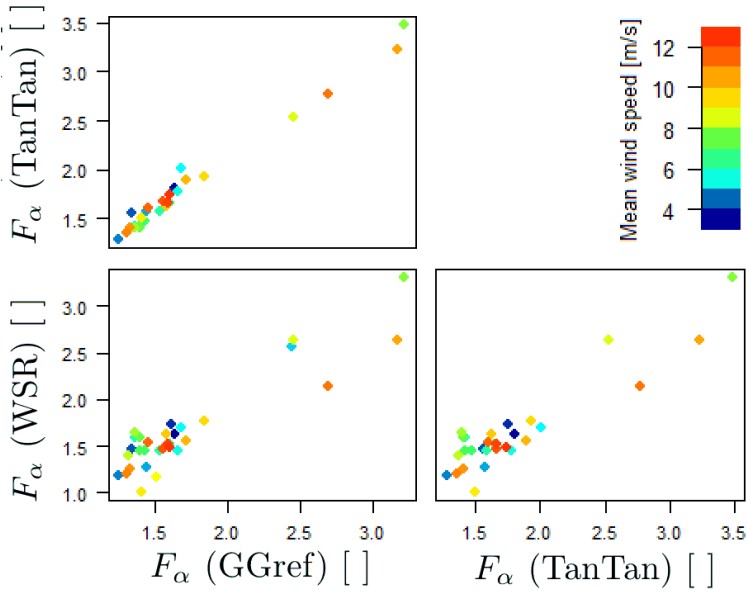

**Figure 9.** $F_\alpha$ calculated with three methods over a large database of wind turbines. Colour coded with the mean wind speed.

# 8 Conclusions

The article presented a new method to calibrate spinner anemometer flow angle measurements (yaw misalignment). The advantage of the method is that it does not need the yaw position of the nacelle to be measured.

The robustness of the method was investigated by repeating the calibration test on the same turbine several times, with the rotor locked in the exact same rotor position to avoid sensor mounting deviations to play a role. The $F_\alpha$ values found for 4 tests for the exact same rotor position were within $\pm 2.7\%$ of the mean value.

The quality score parameter (QSC) was introduced to quantify goodness of the $F_\alpha$ estimate. The QSC was found inversely dependent on the turbulence intensity. To have a sharp estimate of $F_\alpha$ it is therefore better to perform the test in low turbulence wind conditions. The relation found between the QSC and the width of yawing suggests to yaw the turbine further than $\pm 60°$, up to $\pm 80°$ (this values might be different for other spinner shapes). Another issue to consider is that the test could start with an offset, and end up being -90° to 70° instead of -80° to 80°. This is easily avoidable yawing the wind turbine a bit further than the desired yawing span.

The sensitivity of the method to the width of yawing the turbine in and out of the wind was investigated by applying the calibration method to a subset of the original database. The subset was obtained filtering for $\gamma_{ref} \in [-s, s]$, where $s$ was the span ranging from 10° to 90° in steps of 5°. Significant variations of the $F_\alpha$ value were found for yawing span $s$ below approximately 60°.

The $F_\alpha$ calculated with the wind speed response method was compared with the $F_\alpha$ calculated with previous methods (GGref, TanTan) using 29 calibration tests performed by Romo Wind A/S on 17 wind turbine models. The sensitivity to span of yawing showed that the WSR method tends to stabilize to same values as GGref for yawing span larger than approximately 50°. Both the GGref and TanTan methods gave similar values up to $\pm 40°$, then the TanTan method gave higher $F_\alpha$ and diverged for a yawing span larger than 70°.

A recommended yawing span to use to calculate $F_\alpha$ seems to be $\pm 60°$ for the WSR method and $\pm 40°$ for the TanTan and GGref methods, however the turbine shall be yawed further than this angle ($\pm 90°$ recommended) to compensate for initial offset error in the yaw position and wind speed direction change during the test.

It is best to perform the test at the lowest possible turbulence intensity, which might be found in stable atmospheric condition (typically by night) in a flat site with low roughness.

It is recommended to verify the variation of $F_\alpha$ as a function of span of yawing (using the calibrated yaw misalignment if the yaw sensor is not available), since substantially different spinner shapes might give a stable $F_\alpha$ at different yawing span.

*Acknowledgements.* Romo wind A/S for financing one third of the PhD project this article is part of.

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
