# Peer review of "An innovative method to calibrate a spinner anemometer without use of yaw position sensor"

_Wind Energy Science, 2016_

## Referee Comment (RC1) · Anonymous Referee #1 · 9 May 2016

General comments:

This paper is about a new calibration method for spinner anemometers. A spinner anemometer is a combination of three sonic anemometer mounted on the spinner surface of a wind turbine's spinner to measure the horizontal wind speed, the yaw misalignment and the flow inclination angle. T.F. Pedesen, G. Demurtas and F. Zahal (2015) [Wind Energy, 2015, 18:1933-1952] have already developed 5 methods to calibrated a spinner anemometer with however a need of a yaw measurement as a reference. The present calibration methods do not need a reference yaw position sensor, which is not always available in field measurements.

[Figure]

Therefore, the contribution of the authors is of interest for practical implementation of spinner anemometers. The new methods, which consist in keeping a linear relationship between the horizontal wind speed, $U_{hor}$, and the yaw misalignment angle ,$\gamma$ , using adjustment of the calibration coefficient $F_\alpha$, is relatively simple but certainly extracted from a long experience on spinner anemometers from authors. The validation was performed for a large field measurements data-set, on real wind turbines, which allows a sensibility analysis and demonstrates its feasibility.

Major issues:

However, there is no sufficient informations given by the authors to evaluate the method on the present paper. The introduction of the present work is particularly small and poor. Indeed, the main objective of the paper is to adjust the calibration coefficient $F_\alpha$ using a linear "a priori" on the evolution of $U_{hor}$ with $\gamma$. However, the introduction is so short that we don't have the basic relationship between the calibration coefficient and the measured sonic velocity (and other useful definitions, see detailed questions). Also, the example given to demonstrate the method is simplified (yaw misalignment equals the inflow angle), what is the influence of the tilt angle and the flow inclination ? At last, equation 2 (p3, L8), which is the heart of the method, do not express what is written in L6 and L7 (where is $U_{hor,d}$ ?).

The global presentation quality of figures and especially notations of ordinates and abscissa are of poor quality (please match notation in the text).

For these above reasons I recommend the paper after major revisions.

See attached file for detailed review.

Please also note the supplement to this comment:
http://www.wind-energ-sci-discuss.net/wes-2016-10/wes-2016-10-RC1-
supplement.pdf

—————————————————

[Figure]

**Supplement:**

**Submission Date** 2016-05-09
**Date Due** 2016-05-31

**Title «**  An innovative method to calibrate a spinner anemometer without
use of yaw position sensor »

**Corresponding Author** Giorgio Demurtas **Contributing Author**  Nick Gerardus Cornelis Janssen

**General comments:**

This paper is about a new calibration method for spinner anemometers. A spinner anemometer is a combination of three sonic anemometer mounted on the spinner surface of  a wind turbine's spinner to measure the horizontal wind speed, the yaw misalignment and the flow inclination angle. T.F. Pedesen, G. Demurtas and F. Zahal (2015) have already developed 5 methods to calibrated a spinner anemometer with however a need of a yaw measurement as a reference. The present calibration methods do not need a reference yaw position sensor, which is not always available in field measurements.

Therefore, the contribution of the authors is of interest for practical implementation of spinner anemometers. The new methods, which consist in keeping a linear relationship between the horizontal wind speed, $U_{hor}$, and the yaw misalignment angle ,$\gamma$ , using adjustment of the calibration coefficient $F\alpha$, is relatively simple but certainly extracted from a long experience on spinner anemometers from authors. The validation was performed for a large field measurements data-set, on real wind turbines, which allows a sensibility analysis and demonstrates its feasibility.

**Major issues:**

However, there is no sufficient informations given by the authors to evaluate the method on the present paper. The introduction of the present work is particularly small and poor. Indeed, the main objective of the paper is to adjust the calibration coefficient $F\alpha$  using a linear "a priori" on the evolution of $U_{hor}$ with $\gamma$.
However, the introduction is so short that we don't have the basic relationship between the calibration coefficient and the measured sonic velocity (and other useful definitions, see detailed questions).
Also, the example given to demonstrate the method is simplified (yaw misalignment equals the inflow angle), what is the influence of the tilt angle and the flow inclination ?
At last, equation 2 (p3, L8), which is the heart of the method, do not express what is written in L6 and L7 (where is $U_{hor,d}$ ?).

The global presentation quality of figures and especially notations of ordinates and abscissa are of poor quality (please match notation in the text).

For these above reasons I recommend the paper after major revisions.

**See questions below for detailed:**

- The introduction is too short for a person who is not familiar with spinner anemometers. Here is a list of missing informations that I suggest to add:
    1. the relation between wind speeds from sonic anemometer the spinner anemometer coefficients, with coefficients k1 and k2
    2. A figure that defines all angles: tilt angle, yaw angle, flow inclination angle, inflow angle

3. A definition of $K\alpha$ relatively to k1 and k2
4. A definition of the correction factors F1 and F2
5. A definition of $F\alpha$ using F1 and F2
6. Give a relation between the yaw misalignment and $F\alpha$
7. Explain why you need to have a first default value of $K\alpha,d$ before adjusting it using $F\alpha$
8. Give that you used in your model (figure 1) or explain

- p1 L3: "default settings": what are they ?

- p1 L17: replace $K\alpha$ by $K\alpha,d$

- p2 L12: To test the new calibration methods, the authors have tested it on an "artificial data-set". Please explain your data-set: is it CDF computation, wind tunnel measurements, field measurements … ?
- p2 L15: what is "the model" you are talking about ? Give equations of the model (in particular the relation between $U_{hor,d}/U_{hor}$ and $\gamma$)

- p3 L5: in your example (figure 1), you cannot talk about "small inflow angles" but rather about "yaw misalignment angle" unless you have a linear relationship between them ?

- p8 L5: Authors analyze that QSC increase with the wind speed, while in figure 7 we see rather a spreading of QSC with the wind speed. How do you explain that ?

All figures should be improved to match notations in the article (for example figure 1: Uhord → $U_{hor,d}$)

Comment: It would be interesting to look at the boundary layer characteristics (stability...) during the calibration to see how it influences it. Have you looked at it ?

**Minor Issues:**

p1 L3: "measured" appears twice

p4 Figure 2D: you write that the yawing span is +/- 10 and +/- 90° but we don't see any measurements over +/- 70° … what is right, the curve or the text ?

P10 L10: Fa should be replaced by $F\alpha$

---

## Referee Comment (RC2) · Anonymous Referee #2 · 8 Jun 2016

General comments:

The authors propose a method to calibrate spinner anemometers in case a yaw position sensor is not available. The work is of practical interest as typically in order to calibrate an anemometer one has to know the yaw misalignment of the turbine, which is generally difficult to obtain.

The authors use simulated and real field data to demonstrate the effectiveness of the proposed approach. The calibration exploits the fact that the wind speed measure will depend on the yaw misalignment if the anemometer is not calibrated for correctly compensating it. Consequently, the independency of the wind speed measure from the

yaw misalignment is imposed as a tuning constraint.

Even though the idea seems interesting, the paper cannot be accepted in the current status since there are technical issues to be solved/clarified, missing descriptions which may ease the comprehension and the evaluation of the work, incongruences in the notation and unclear plots.

Personally, I recommend the paper for publication after major revisions. Specific comments and minor corrections are listed below.

Specific comments:

1. It is mentioned that the spinner anemometer gives three outputs, wind speed, yaw angle and upflow angle. The upflow angle is however not considered in the treatment. Is it possible to comment this point and possibly to discuss the effects of upflow angle changes on the calibration results?

2. From equation (1) it seems that the correction consists only in a scale factor. Is it possible to explain why a bias was not considered?

3. What is $\kappa_{\alpha,d}$ (default factor)? and how is it computed?

4. During the calibration, the wind speed is assumed constant. This seems a really stringent requirement especially when the tests last one hour or more (as in this paper). The author should comment on this.

5. From the sensitivity analysis (Sec. 5), the fact that the method be sensitive to the used misalignment range seems clear. In fact, when the turbine is yawed in a range $\pm 60$ deg the scale factor results equal to more or less 1.67, whereas to about 1.5 for a range of $\pm 90$. Please comment.
6. Figure 5 is displayed but not mentioned in the text.

7. Figures 7 and 8 are unclear. x- and y-label may help one understand. Better explanations in the text are also mandatory.

Minor corrections

1. Pag. 1, line 15: please, add definitions for all symbols ($U_{hor}$, $\gamma$, $\beta$), which appear here for the very first time.

2. Pag. 1, line 17 and equation (1): please, add the definition of $\kappa_{\alpha,d}$

3. Pag. 2, fig. 1: $U_{hord}$ in the ylabel is displayed without description.

4. Pag. 3, line 6: "consists"

5. Pag. 3, eq. (2): definitions of symbols missing

6. Pag. 3, line 10: please provide a reference for the optimization method.

7. Pag. 4, fig. 2: units of measurements of the plots missing

8. Pag. 5, fig. 3: units of measurements of the plots missing

9. Pag. 6, line 13: $\gamma_{ref}$ appears here for the first time without explanation

10. Pag. 6, tab. 1: in the table $F_a$ should be substituted with $F_\alpha$

11. Pag. 7, fig. 5: in the figure $F_a$ should be substituted with $F_\alpha$

12. Pag. 8, line 15: "increases"

13. Pag. 9, fig. 7: color code missing
14. Pag. 10, fig. 8: color code missing

---

## Referee Comment (RC3) · Anonymous Referee #3 · 13 Jun 2016

General comments

The paper presents an alternative method of calibrating spinner anemometers. A spinner anemometer is a custom implementation of sonic anemometry, with 3 1D sensors individually mounted on a wind turbine' spinner surface. Horizontal wind speed, yaw misalignment and flow inclination angle are obtained from raw signals (wind velocity at sensor) processed based on spinner and blade root geometry as well as sensor location on spinner.

As opposed to 2 other previously developed calibration procedures, the proposed method does not require the availability of turbine yaw position information. Since this scenario is encountered in many wind turbines, the proposed method is of practical interest. The new method is based on observed dependency between yaw misalignment and the wind speed measurement from a spinner anemometer. Combining this observation with simulated data and field measurements acquired in constant wind speed conditions (over specific yawing span), the authors define a method of optimizing the correction factor necessary to adjust the "default" calibration value in order to obtain a "correct" calibration value.

While the implications of the new calibration method are evident and the authors present a good amount of data to support some of their observations and reasoning, the paper does not supply enough information to validate key assumptions and the quality of results. The overall formatting of the paper is not optimal with multiple notation inconsistencies and incomplete graphics obstructing comprehension.

Based on the above observations, I recommend this paper for publication after major revisions. Suggested corrections attached below.

Major corrections

- [section 5, 6, 8] The proposed calibration method is evaluated based on the resulted correction factor and related spinner anemometer data. While this offers a measure of reproducibility, it does not imply final result accuracy (especially in the absence of reference measurements validating the initial assumptions). Comparison of spinner anemometer data versus reference measurements should be included as a measure of effectiveness for the new calibration method.

- [section 3] Given the strict assumption of "constant wind speed", zero flow inclination and the lengthy measurement/yawing procedure, site reference measurements (wind speed, direction, turbulence obtained from another instrument, derived yaw misalignment and flow inclination angle) should be provided

- [section 5] Tests 1, 2, 4 and 6 were discarded from the final calibration factor variation. The reasoning (is it yawing span or different rotor positioning?) should be elaborated

given that the resulting ±2.7% variation represents a major evaluation of the method and the related ±90° yawing span (tests 7-10) contradicts the final recommendation of ±60° [P11 L14].

- Given the current form of this paper, the referenced "Calibration of a spinner anemometer for flow angle measurements" is a must-read. A minimum necessary amount of background information (description of symbols, coordinate systems, etc.) should be included in the paper.

- All notations, plot labeling, plot legends and figure descriptions should be checked for consistency and completeness.

Minor corrections

- [P2 L14] The assumption that yaw misalignment equals inflow angle should be augmented by a short explanation on why this will not affect the efficiency of the calibration method and/or results accuracy.

- [P10 L2] The statement "a certain level of agreement between the three methods" could use further elaboration, especially given the evident agreement between GGref and TanTan.

- [figure 2D] Data for 75 to 90 deg angles missing

- [figure 2,3] Instances of correction factor related to wind speed calibration should be eliminated or explained accordingly.

---

## Author Comment (AC1) · 29 Jun 2016

thanks for your comments, I considered them all and included in a reviewed manuscript which is ready. BR Giorgio

---

## Author Response (AR1)

General comments:

This paper is about a new calibration method for spinner anemometers. A spinner anemometer is a combination of three sonic anemometer mounted on the spinner surface of a wind turbine's spinner to measure the horizontal wind speed, the yaw misalignment and the flow inclination angle. T.F. Pedesen, G. Demurtas and F. Zahal (2015) [Wind Energy, 2015, 18:1933-1952] have already developed 5 methods to calibrated a spinner anemometer with however a need of a yaw measurement as a reference. The present calibration methods do not need a reference yaw position sensor, which is not always available in field measurements.

*introduce the conversion box connection to sensors and purpose*

Therefore, the contribution of the authors is of interest for practical implementation of spinner anemometers. The new methods, which consist in keeping a linear relationship between the horizontal wind speed, $U_{hor}$, and the yaw misalignment angle , $\gamma$ , using adjustment of the calibration coefficient $F_\alpha$, is relatively simple but certainly extracted from a long experience on spinner anemometers from authors. The validation was performed for a large field measurements data-set, on real wind turbines, which allows a sensibility analysis and demonstrates its feasibility.

Major issues:

However, there is no sufficient informations given by the authors to evaluate the method on the present paper. The introduction of the present work is particularly small and poor. Indeed, the main objective of the paper is to adjust the calibration coefficient $F_\alpha$ using a linear "a priori" on the evolution of $U_{hor}$ with $\gamma$. However, the introduction is so short that we don't have the basic relationship between the calibration coefficient and the measured sonic velocity (and other useful definitions, see detailed questions). Also, the example given to demonstrate the method is simplified (yaw misalignment equals the inflow angle), what is the influence of the tilt angle and the flow inclination ? At last, equation 2 (p3, L8), which is the heart of the method, do not express what is written in L6 and L7 (where is $U_{hor,d}$ ?).

The global presentation quality of figures and especially notations of ordinates and abscissa are of poor quality (please match notation in the text).

For these above reasons I recommend the paper after major revisions.

See attached file for detailed review.

Please also note the supplement to this comment:
http://www.wind-energ-sci-discuss.net/wes-2016-10/wes-2016-10-RC1-
supplement.pdf

$\beta$ is measured and $\delta$ is set into the spinner anemometer
conversion box. The spinner anemometer output $\delta$ (1)
is therefore already purified from the effect of those
two. variables.

**Submission Date** 2016-05-09
**Date Due** 2016-05-31

**Title** «  An innovative method to calibrate a spinner anemometer without use of yaw position sensor »

**Corresponding Author** Giorgio Demurtas **Contributing Author**  Nick Gerardus Cornelis Janssen

**General comments:**

This paper is about a new calibration method for spinner anemometers. A spinner anemometer is a combination of three sonic anemometer mounted on the spinner surface of  a wind turbine's spinner to measure the horizontal wind speed, the yaw misalignment and the flow inclination angle. T.F. Pedesen, G. Demurtas and F. Zahal (2015) have already developed 5 methods to calibrated a spinner anemometer with however a need of a yaw measurement as a reference. The present calibration methods do not need a reference yaw position sensor, which is not always available in field measurements.

Therefore, the contribution of the authors is of interest for practical implementation of spinner anemometers. The new methods, which consist in keeping a linear relationship between the horizontal wind speed, $U_{hor}$, and the yaw misalignment angle ,$\gamma$ , using adjustment of the calibration coefficient $F\alpha$, is relatively simple but certainly extracted from a long experience on spinner anemometers from authors. The validation was performed for a large field measurements data-set, on real wind turbines, which allows a sensibility analysis and demonstrates its feasibility.

**Major issues:**

However, there is no sufficient informations given by the authors to evaluate the method on the present paper. The introduction of the present work is particularly small and poor. Indeed, the main objective of the paper is to adjust the calibration coefficient $F\alpha$  using a linear "a priori" on the evolution of $U_{hor}$ with $\gamma$.
However, the introduction is so short that we don't have the basic relationship between the calibration coefficient and the measured sonic velocity (and other useful definitions, see detailed questions).
Also, the example given to demonstrate the method is simplified (yaw misalignment equals the inflow angle), what is the influence of the tilt angle and the flow inclination ?
At last, equation 2 (p3, L8), which is the heart of the method, do not express what is written in L6 and L7 (where is $U_{hor,d}$ ?).

The global presentation quality of figures and especially notations of ordinates and abscissa are of poor quality (please match notation in the text).

For these above reasons I recommend the paper after major revisions.

**See questions below for detailed:**

- The introduction is too short for a person who is not familiar with spinner anemometers. Here is a list of missing informations that I suggest to add:
  1.  the relation between wind speeds from sonic anemometer the spinner anemometer coefficients, with coefficients k1 and k2
  2.  A figure that defines all angles: tilt angle, yaw angle, flow inclination angle, inflow angle

✓ 3. A definition of Kα relatively to k1 and k2

GR6 ✓ 4. A definition of the correction factors F1 and F2

✗✓ 5. A definition of Fα using F1 and F2

6. Give a relation between the yaw misalignment and Fα  *Fα is a constant. Added relation with Kβ*

✓ 7. Explain why you need to have a first default value of Kα,d before adjusting it using Fα

✓ 8. Give that you used in your model (figure 1) or explain *eq.*

✓ - p1 L3: "default settings": what are they ?

✓ - p1 L17: replace Kα by Kα,d

*S WRITTEN*
*✓ P2 L13* ✓ - p2 L12: To test the new calibration methods, the authors have tested it on an "artificial data-set". Please explain your data-set: is it CDF computation, wind tunnel measurements, field measurements ...? *REORGANIZED COMPLETELY*

✓ - p2 L15: what is "the model" you are talking about ? Give equations of the model (in particular the relation between Uhor,d/Uhor and γ)

✓ - p3 L5: in your example (figure 1), you cannot talk about "small inflow angles" but rather about "yaw misalignment angle" unless you have a linear relationship between them ? *SEE*
*Is THE SAME. α=γ, see P2 L13, when β=0*

- p8 L5: Authors analyze that QSC increase with the wind speed, while in figure 7 we see rather a spreading of QSC with the wind speed. How do you explain that ? *TRUE.*
*High wind speed can't guarantee high QSC, while low TI can.*

✓ All figures should be improved to match notations in the article (for example figure 1: Uhord → Uhor,d)

✓ Comment: It would be interesting to look at the boundary layer characteristics (stability...) during the calibration to see how it influences it. Have you looked at it ? *NO BECAUSE I'VE NO DATA BUT SURELY LOW T.I (STABLE COND) WOULD HELP*

**Minor Issues:**

✓ p1 L3: "measured" appears twice

*Chap 3*
*Added text* ✓ p4 Figure 2D: you write that the yawing span is +/- 10 and +/- 90° but we don't see any measurements over +/- 70° ... what is right, the curve or the text ? *WSR Kβ is calculated only if the amount of measurement in the subset 5° of the range is >30"*

✓ P10 L10: Fa should be replaced by Fα

Wind Energ. Sci. Discuss.,
doi:10.5194/wes-2016-10-RC2, 2016

[Figure]

WIND
ENERGY
SCIENCE
DISCUSSIONS

The authors propose a method to calibrate spinner anemometers in case a yaw position sensor is not available. The work is of practical interest as typically in order to calibrate an anemometer one has to know the yaw misalignment of the turbine, which is generally difficult to obtain.

The authors use simulated and real field data to demonstrate the effectiveness of the proposed approach. The calibration exploits the fact that the wind speed measure will depend on the yaw misalignment if the anemometer is not calibrated for correctly compensating it. Consequently, the independency of the wind speed measure from the

yaw misalignment is imposed as a tuning constraint.

Even though the idea seems interesting, the paper cannot be accepted in the current status since there are technical issues to be solved/clarified, missing descriptions which may ease the comprehension and the evaluation of the work, incongruences in the notation and unclear plots.

Personally, I recommend the paper for publication after major revisions. Specific comments and minor corrections are listed below.

Specific comments:

1. It is mentioned that the spinner anemometer gives three outputs, wind speed, yaw angle and upflow angle. The upflow angle is however not considered in the treatment. Is it possible to comment this point and possibly to discuss the effects of upflow angle changes on the calibration results?

2. From equation (1) it seems that the correction consists only in a scale factor. Is it possible to explain why a bias was not considered? *see eg* $V_1, V_2, V_3$

3. What is $\kappa_{\alpha,d}$ (default factor)? and how is it computed? $=1$

4. During the calibration, the wind speed is assumed constant. This seems a really stringent requirement especially when the tests last one hour or more (as in this paper). The author should comment on this. *ADDED IN DISCUSSION*

5. From the sensitivity analysis (Sec. 5), the fact that the method be sensitive to the used misalignment range seems clear. In fact, when the turbine is yawed in a range ±60 deg the scale factor results equal to more or less 1.67, whereas to about 1.5 for a range of ±90. Please comment. *ADDED IN SENS. ANALYS*

✓ 6. Figure 5 is displayed but not mentioned in the text.

7. Figures 7 and 8 are unclear. x- and y-label may help one understand. Better explanations in the text are also mandatory.

Minor corrections

✓ 1. Pag. 1, line 15: please, add definitions for all symbols ($U_{hor}$, $\gamma$, $\beta$), which appear here for the very first time.

✓ 2. Pag. 1, line 17 and equation (1): please, add the definition of $\kappa_{\alpha,d}$

∿ 3. Pag. 2, fig. 1: $U_{hord}$ in the ylabel is displayed without description.

✓ 4. Pag. 3, line 6: "consists"

*NEW FIG LIST* — 5. Pag. 3, eq. (2): definitions of symbols missing

✓ 6. Pag. 3, line 10: please provide a reference for the optimization method.

✓ 7. Pag. 4, fig. 2: units of measurements of the plots missing

✓ 8. Pag. 5, fig. 3: units of measurements of the plots missing    *YAWING SPAN*

✓ 9. Pag. 6, line 13: $\gamma_{ref}$ appears here for the first time without explanation   *SEC 4.4*

✓ 10. Pag. 6, tab. 1: in the table $F_a$ should be substituted with $F_\alpha$

✓ 11. Pag. 7, fig. 5: in the figure $F_a$ should be substituted with $F_\alpha$

✓ 12. Pag. 8, line 15: "increases"

✓ 13. Pag. 9, fig. 7: color code missing

✓ 14. Pag. 10, fig. 8: color code missing

Wind Energ. Sci. Discuss.,
doi:10.5194/wes-2016-10-RC3, 2016

[Figure]

**WIND ENERGY SCIENCE DISCUSSIONS**

The paper presents an alternative method of calibrating spinner anemometers. A spinner anemometer is a custom implementation of sonic anemometry, with 3 1D sensors individually mounted on a wind turbine' spinner surface. Horizontal wind speed, yaw misalignment and flow inclination angle are obtained from raw signals (wind velocity at sensor) processed based on spinner and blade root geometry as well as sensor location on spinner.

As opposed to 2 other previously developed calibration procedures, the proposed method does not require the availability of turbine yaw position information. Since this scenario is encountered in many wind turbines, the proposed method is of practical interest. The new method is based on observed dependency between yaw misalignment and the wind speed measurement from a spinner anemometer. Combining this observation with simulated data and field measurements acquired in constant wind speed conditions (over specific yawing span), the authors define a method of optimizing the correction factor necessary to adjust the "default" calibration value in order to obtain a "correct" calibration value.

While the implications of the new calibration method are evident and the authors present a good amount of data to support some of their observations and reasoning, the paper does not supply enough information to validate key assumptions and the quality of results. The overall formatting of the paper is not optimal with multiple notation inconsistencies and incomplete graphics obstructing comprehension.

Based on the above observations, I recommend this paper for publication after major revisions. Suggested corrections attached below.

Major corrections

- [section 5, 6, 8] The proposed calibration method is evaluated based on the resulted correction factor and related spinner anemometer data. While this offers a measure of reproducibility, it does not imply final result accuracy (especially in the absence of reference measurements validating the initial assumptions). Comparison of spinner anemometer data versus reference measurements should be included as a measure of effectiveness for the new calibration method. \ hoa !

*CAN ONLY IN WIND TUNNEL*

- [section 3] Given the strict assumption of "constant wind speed", zero flow inclination and the lengthy measurement/yawing procedure, site reference measurements (wind speed, direction, turbulence obtained from another instrument, derived yaw misalignment and flow inclination angle) should be provided

*IS NOT A PROBLEM BECAUSE I GET γ*

- [section 5] Tests 1, 2, 4 and 6 were discarded from the final calibration factor variation. The reasoning (is it yawing span or different rotor positioning?) should be elaborated

*BECAUSE THE ±2,7%. IS DUE TO THE METHOD*

*WHILE THE 8,5 % INCLUDES SPINNER A. DISSIMETRIES (DIFFERENT ROTOR POSITION)*

*CONTROLLER DISPLAY*

given that the resulting ±2.7% variation represents a major evaluation of the method and the related ±90° yawing span (tests 7-10) contradicts the final recommendation of ±60° [P11 L14].

*ANALYSIS*

- Given the current form of this paper, the referenced "Calibration of a spinner anemometer for flow angle measurements" is a must-read. A minimum necessary amount of background information (description of symbols, coordinate systems, etc.) should be included in the paper.

- All notations, plot labeling, plot legends and figure descriptions should be checked for consistency and completeness.

Minor corrections

- [P2 L14] The assumption that yaw misalignment equals inflow angle should be augmented by a short explanation on why this will not affect the efficiency of the calibration method and/or results accuracy.

- [P10 L2] The statement "a certain level of agreement between the three methods" could use further elaboration, especially given the evident agreement between GGref and TanTan.

- [figure 2D] Data for 75 to 90 deg angles missing — *BECAUSE <30" OF MEASUREMENTS*

- [figure 2,3] Instances of correction factor related to wind speed calibration should be eliminated or explained accordingly.

OK